# Mechanism of stepwise electron transfer in six-transmembrane epithelial antigen of the prostate (STEAP) 1 and 2

Kehan Chen[1†], Lie Wang[1†], Jiemin Shen[1], Ah-Lim Tsai[2], Ming Zhou[1*], Gang Wu[2*]

[1]Verna and Marrs McLean Department of Biochemistry and Molecular Pharmacology, Baylor College of Medicine, Houston, United States; [2]Division of Hematology-Oncology, Department of Internal Medicine, University of Texas-McGovern Medical School, Houston, United States

**Abstract** Six transmembrane epithelial antigen of the prostate (STEAP) 1–4 are membrane-embedded hemoproteins that chelate a heme prosthetic group in a transmembrane domain (TMD). STEAP2–4, but not STEAP1, have an intracellular oxidoreductase domain (OxRD) and can mediate cross-membrane electron transfer from NADPH via FAD and heme. However, it is unknown whether STEAP1 can establish a physiologically relevant electron transfer chain. Here, we show that STEAP1 can be reduced by reduced FAD or soluble cytochrome $b_5$ reductase that serves as a surrogate OxRD, providing the first evidence that STEAP1 can support a cross-membrane electron transfer chain. It is not clear whether FAD, which relays electrons from NADPH in OxRD to heme in TMD, remains constantly bound to the STEAPs. We found that FAD reduced by STEAP2 can be utilized by STEAP1, suggesting that FAD is diffusible rather than staying bound to STEAP2. We determined the structure of human STEAP2 in complex with $NADP^+$ and FAD to an overall resolution of 3.2 Å by cryo-electron microscopy and found that the two cofactors bind STEAP2 similarly as in STEAP4, suggesting that a diffusible FAD is a general feature of the electron transfer mechanism in the STEAPs. We also demonstrated that STEAP2 reduces ferric nitrilotriacetic acid ($Fe^{3+}$-NTA) significantly slower than STEAP1 and proposed that the slower reduction is due to the poor $Fe^{3+}$-NTA binding to the highly flexible extracellular region in STEAP2. These results establish a solid foundation for understanding the function and mechanisms of the STEAPs.

## eLife assessment

This study provides **useful** insights into the mechanisms of electron transport in STEAP proteins, consistent with current models. The work strengthens and supports previously published biochemical and structural data, and the experimental results are of **solid** technical quality. The manuscript will be of interest to colleagues who work on STEAP proteins and related electron transfer systems.

## Introduction

Six-transmembrane epithelial antigen of the prostate 1 (STEAP1) was first discovered owing to its high level of expression in prostate cancer cells (**Hubert et al., 1999**). Discovery of STEAP2–4 soon followed, and further analyses show that STEAP2–4 have metal ion reductase activities (**Grunewald et al., 2012**; **Ohgami et al., 2006**). STEAP3 was identified as a ferrireductase required for iron uptake in erythroid cells (**Ohgami et al., 2006**; **Ohgami et al., 2005**). STEAP2–4 are also found overexpressed in many types of cancer cells, suggesting their involvement in cancer initiation or progression (**Hubert et al., 1999**; **Gomes et al., 2013**; **Rocha et al., 2021**).

**\*For correspondence:**
mzhou@bcm.edu (MZ);
gang.wu@uth.tmc.edu (GW)

[†]These authors contributed equally to this work

**Competing interest:** The authors declare that no competing interests exist.

Each STEAP protein has a transmembrane domain (TMD) which consists of six transmembrane helices (TM), and STEAP2–4, but not STEAP1, also have an N-terminal intracellular oxidoreductase domain (OxRD) (*Pick, 2023*; *Kleven et al., 2015*). TMD of the STEAP family of proteins are homologous to membrane-embedded reductases including mammalian NADPH oxidases (NOX) and dual oxidases (DUOX), and bacterial and yeast ferric reductases (*Zhang et al., 2013*). The structures of NOX and DUOX show that their TMD binds two heme prosthetic groups, each ligated to a pair of conserved histidine residues from TMs 3 and 5; one heme is close to the intracellular side and the other close to the extracellular side of the TMD (*Wu et al., 2021*; *Sun, 2020*; *Liu et al., 2022*; *Magnani et al., 2017*; *Oosterheert et al., 2020*). In NOX or DUOX, the OxRD binds both NADPH and FAD, and the cross-membrane electron transfer chain starts with hydride transfer from NADPH to FAD, and then sequentially to the intracellular and extracellular hemes, and finally to the substrate. The structure of STEAP4, on the other hand, shows that only one heme is present in the TMD and it corresponds to the extracellular heme in NOX or DUOX (*Oosterheert et al., 2018*). In STEAP4, the FAD straddles OxRD and TMD with the isoalloxazine ring of FAD binding at the equivalent position of the intracellular heme in NOX and DUOX and the nucleotide moiety of FAD binding to the OxRD (*Figure 1*). While this configuration allows electron transfer between FAD and heme, the isoalloxazine ring is too far away from the nicotinamide ring of NADPH to receive hydride. Thus, the isoalloxazine ring of FAD must dissociate from the TMD and move closer to the nicotinamide ring of NADPH for hydride transfer. Although the isoalloxazine ring of FAD must assume different conformations during the redox cycles in STEAPs, its adenosine moiety could stay bound to the OxRD. However, here we show evidence that FAD does not stay tightly bound to the STEAP protein and can become diffusible after its reduction.

STEAP1 has been considered not be able to establish an electron transfer chain due to its lack of an OxRD, however, recent structures of STEAP1 and STEAP4 offer some hints otherwise (*Oosterheert et al., 2018*; *Oosterheert and Gros, 2020*). In the STEAP4 structure, the isoalloxazine ring of FAD is coordinated by residues in TMD that are conserved in STEAP1. The structure of STEAP1 aligns well with that the TMD of STEAP4, and although FAD is not present in the STEAP1 structure (*Oosterheert and Gros, 2020*), we showed previously that purified STEAP1 contains residual FAD, and that FAD binds to STEAP1 with $K_D$ of ~32 µM (*Kim et al., 2016*). Building upon these knowledge, we focused on identifying conditions that allow formation of a transmembrane electron transfer chain in STEAP1. We also determined the structure of human STEAP2, which shows that it has a more flexible substrate binding site compared to STEAP1 and STEAP4.

## Results
### Reduction of STEAP1 by reduced FAD

We first examined if the heme on STEAP1 can be reduced by FAD. We found that reduced FAD (as $FADH^-$) readily reduces STEAP1, shown by the fast decrease in the Soret absorbance of ferric heme ($A_{413}$) and the concomitant increase in the Soret absorbance of ferrous heme ($A_{427}$) (*Figure 2A*). The time course of $A_{427}$ increase is biphasic. The fast phase has a rate constant of 7.7 s$^{-1}$ and accounts for 60% of the total change at $A_{427}$, and the slow phase, 0.67 s$^{-1}$ and 40% (*Figure 2B*). The rate constants of both phases exhibit dependence on [$FADH^-$]. The $V_{max}$ and $K_M$ are estimated 12 s$^{-1}$ and 4.7 µM for the fast phase, and the parameters are estimated 0.9 s$^{-1}$ and 2.7 µM for the slow phase (*Figure 2B*, inset).

In STEAP4 structure, a phenylalanine side chain (Phe359) is positioned between the isoalloxazine ring of FAD and the heme and is thought to mediate electron transfer from FAD to heme (*Oosterheert et al., 2018*). In STEAP1–3, the equivalent residue is a leucine, Leu230 in STEAP1 (*Figure 2—figure supplement 1A*). To examine if Leu230 is involved in electron transfer, we expressed and purified the L230G STEAP1 mutant. The UV-Vis spectrum of L230G STEAP1 is identical to that of the wild-type (WT) STEAP1 (*Figure 2—figure supplement 1B*), indicating that the mutation does not perturb heme binding. When $FADH^-$ was added to L230G STEAP1, biphasic reduction of heme was observed and the $k_{obs}$'s showed dependence on [$FADH^-$]. The $V_{max}$ and $K_M$ are estimated 2 s$^{-1}$ and 3.6 µM for the fast phase, and the parameters are estimated 0.16 s$^{-1}$ and 1.1 µM for the slow phase (*Figure 2—figure supplement 1C*). These results indicate that the heme in L230G STEAP1 is reduced by $FADH^-$ more

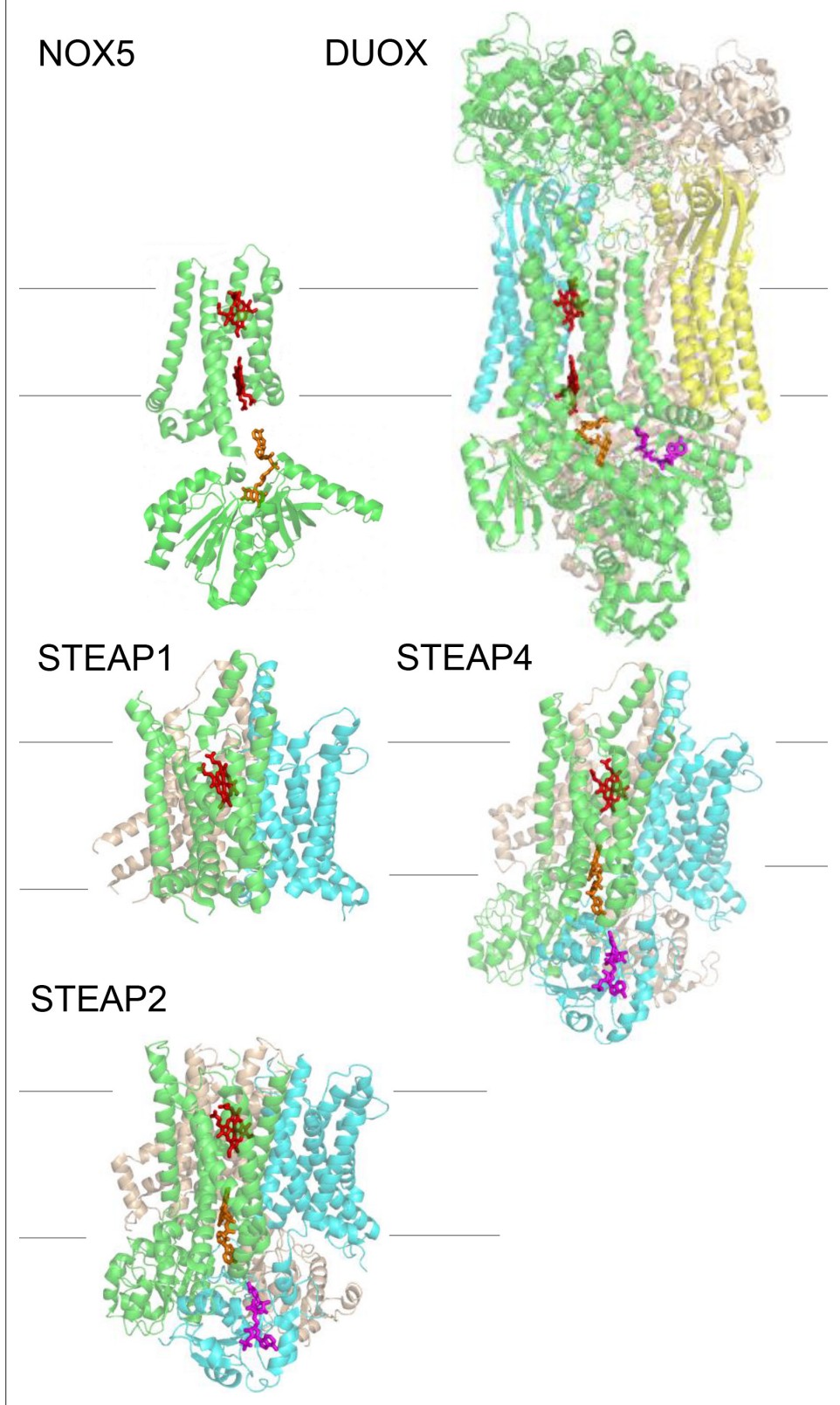

**Figure 1.** The structures of NADPH oxidases (NOX), dual oxidases (DUOX), and six-transmembrane epithelial antigen of the prostate (STEAPs). The crystal structures of the transmembrane domain (TMD) and oxidoreductase domain (OxRD) of NOX5 (from *Cylindrospermum stagnale*) are plotted together (PDB codes: TMD, 5o0t; OxRD, 5o0x). The cryo-electron microscopy (cryo-EM) structure of human DUOX1 (PDB code: 7d3f) shows dimeric

*Figure 1 continued on next page*

*Figure 1 continued*

oligomerization (green and cyan) complexed with DUOX auxiliary protein A1 (DUOXA1, wheat and yellow). The cryo-EM structures of STEAPs are homotrimers (green, cyan, and wheat. PDB codes: STEAP1, 6y9b; STEAP4, 6hcy; STEAP2, 7tai). In DUOX and STEAPs, only one set of cofactors and heme group are shown for clarity. Heme, red; FAD, orange; NADP$^+$, magenta. The lines represent cell membrane (top, extracellular and bottom, intracellular).

than five times slower than in WT STEAP1, suggesting that the side chain of Leu230 is involved in mediating electron transfer between FAD and heme in STEAPs.

## Reduction of STEAP1 by cytochrome $b_5$ reductase

We proceeded to test whether the heme on STEAP1 can be reduced by cytochrome $b_5$ reductase ($b_5$R). $b_5$R catalyzes the reduction of a tightly bound FAD by NADH and is known to reduce the heme on cytochrome $b_5$ (*Hall et al., 2022*; *Shen et al., 2022*). Under anaerobic conditions, STEAP1 does not react with NADH but in the presence of both $b_5$R and NADH, the $A_{427}$ absorbance increases indicating reduction of the heme on STEAP1 (*Figure 3—figure supplement 1A*). Using rapid-scan stopped-flow method, we captured the kinetics of the reactions when STEAP1 pre-incubated with $b_5$R was mixed with NADH (*Figure 3A*). The reduction of STEAP1 is clearly indicated by the shift of Soret peak ($A_{413}$ to $A_{427}$) and the split and increase of the α and β bands at 560 nm and 530 nm, respectively (*Figure 3A*). Three spectral species, $A$, $B$, and $C$, are resolved with rate constants of 177.9 (±35.3) s$^{-1}$ ($A$ to $B$) and 0.13 (±0.006) s$^{-1}$ ($B$ to $C$), respectively (*Figure 3A*, inset). Spectral species $A$ corresponds to ferric STEAP1 plus oxidized $b_5$R while species $C$ is ferrous STEAP1 with fully reduced $b_5$R. A spectral intermediate species $B$ can be identified with decreased absorbance in 420–500 nm range but little or no change in the Soret range when compared to $A$ (*Figure 3A*, inset), indicating that the intermediate $B$ contains partially reduced $b_5$R and a ferric heme. The resolution of intermediate $B$ is due to fast FAD reduction in $b_5$R by NADH without significant STEAP1 reduction. The $B$ to $C$ conversion reflects electron transfer from reduced $b_5$R to STEAP1 (*Figure 3A*), indicating that STEAP1 can form an electron transfer chain with $b_5$R.

We next examined whether $b_5$R forms a complex with STEAP1. Using bio-layer interferometry (BLI) assay, we measured the affinity between $b_5$R and STEAP1 and found that $b_5$R binds STEAP1 with a $K_D$ of ~5.9 μM (*Figure 3—figure supplement 2*). We noticed that the fits to the association and dissociation profiles in BLI assay are rather poor at high concentrations of $b_5$R and this is likely due to non-specific interactions between STEAP1 with $b_5$R. Nonetheless, our BLI data demonstrates that $b_5$R can dock onto STEAP1 to establish an electron transfer chain.

We also examined reduction of L230G STEAP1 by NADH and $b_5$R (*Figure 3B*). Three spectral species, $A$, $B$, and $C$, are resolved, and the rate constants are 78.8 (±22.6) s$^{-1}$ ($A$ to $B$) and 0.02 (±0.01) s$^{-1}$ ($B$ to $C$) (*Figure 3B*, inset). Species $A$ corresponds to ferric L230G STEAP1 with oxidized $b_5$R while species $C$ represents ferrous L230G STEAP1 with fully reduced $b_5$R (*Figure 3B*, inset). As in WT STEAP1, the fast reduction of $b_5$R by NADH leads to the resolution of spectral intermediate $B$, which has decreased absorbance between 420 nm and 500 nm but approximately the same Soret absorbance compared to $A$ (*Figure 3B*, inset). On the other hand, the rate constant from $B$ to $C$, which is electron transfer rate from reduced $b_5$R to the heme, is significantly slower in L230G STEAP1 than in WT STEAP1 (*Figure 3B*, inset), suggesting that Leu230 is involved in the electron transfer from $b_5$R to STEAP1.

## Purification and characterization of STEAP2

While structures of full-length STEAP1 and STEAP4 have been determined by cryo-electron microscopy (cryo-EM) (*Oosterheert et al., 2018*; *Oosterheert and Gros, 2020*), those of full-length STEAP2 and STEAP3 remain unresolved. We expressed and purified human STEAP2, which elutes as a single peak in size exclusion chromatography and the elution volume is consistent with STEAP2 being a homotrimer (*Figure 4A*). A prominent heme absorption peak is present in the purified STEAP2, and the heme content typically ranges from 70% to 90%. FAD is also detected in the purified STEAP2, however, at a level typically lower than 20% based on the fluorescence of FAD released from denatured STEAP2. No NADP(H) is detected in the purified STEAP2, suggesting that its association with STEAP2 is more transient than either heme or FAD.

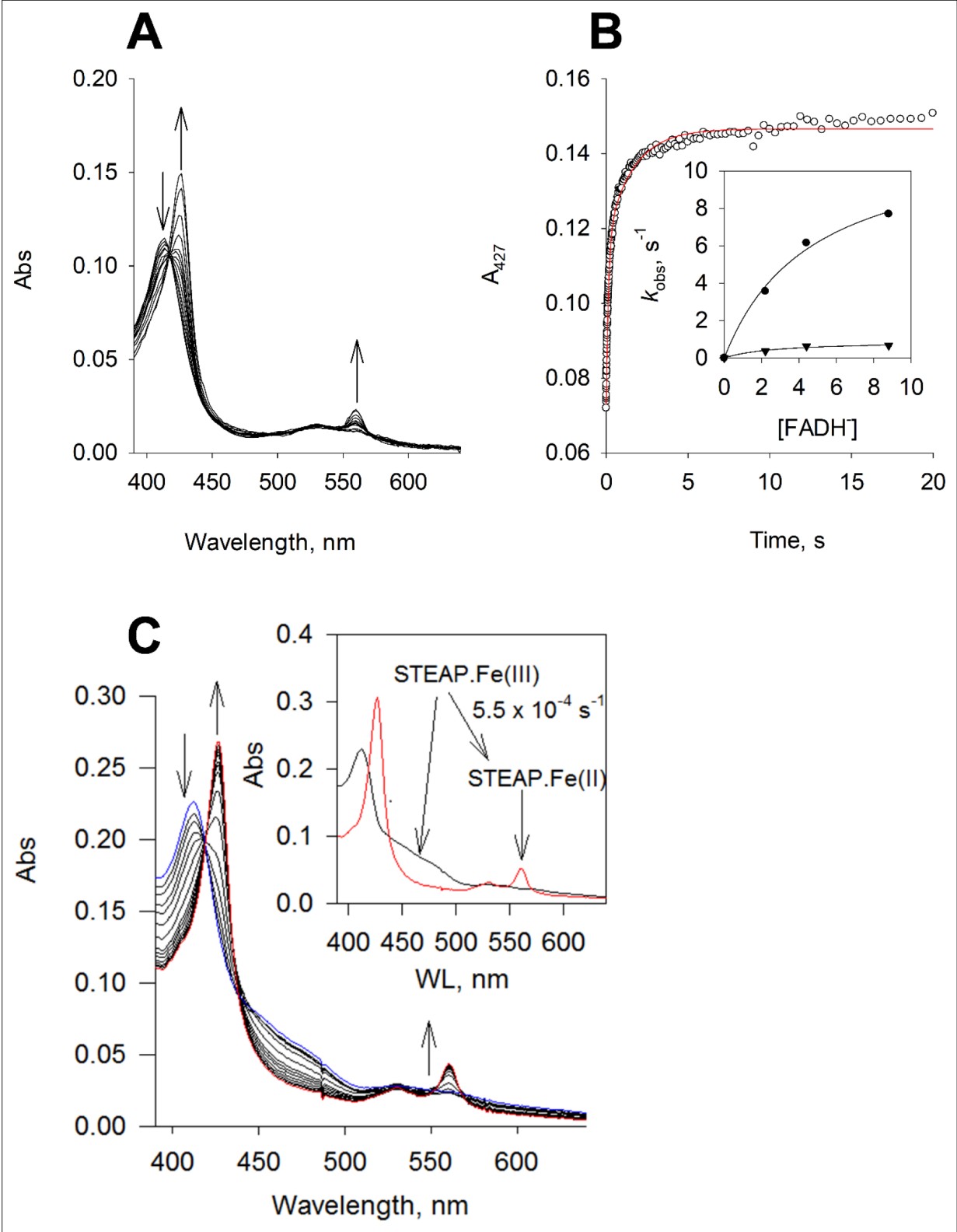

**Figure 2.** Reduction of the heme on six-transmembrane epithelial antigen of the prostate 1 (STEAP1). (**A**) Rapid-scan reaction of 1.1 µM STEAP1 with 4.5 µM reduced FAD (FADH⁻); the spectral change was monitored for 20 s. (**B**) The time course of $A_{427}$ (black), the Soret absorbance of ferrous heme, was extracted from the rapid-scan data. Red: biphasic exponential fit with rate constants $k_{obs}$ of 7.7 (±0.30) and 0.67 (±0.034) s⁻¹, respectively (n=3). The percentage of each phase is 60% and 40%, respectively. Inset, the dependence of rate constants on [FADH⁻]. Dot, the fast phase; triangle, the slow phase. Lines, fit with equation $k_{obs} = V_{max} \cdot [FADH^-]/(K_M + [FADH^-])$. (**C**) The spectral changes in the reaction of a mixture of 1.1 µM STEAP2 and

*Figure 2 continued on next page*

*Figure 2 continued*

0.9 µM STEAP1 (plus 2.2 µM FAD) with 60 µM NADPH; the spectral change was monitored for 1 hr. The direction of the spectral changes is indicated by the arrows. Blue, the spectrum captured at the start of the reaction; red, the spectrum after 1 hr reaction. Inset, the resolved spectral species by deconvolution and the conversion rate constant. Black, ferric STEAP and red, ferrous STEAP.

The online version of this article includes the following figure supplement(s) for figure 2:

**Figure supplement 1.** Electron transfer in six-transmembrane epithelial antigen of the prostate (STEAPs) may be mediated by a bulky side chain.

The UV-Vis spectra of STEAP2 heme are identical to those of STEAP1 heme. Ferric STEAP2 shows a Soret band at 413 nm and a broad Q band centered around 550 nm while the Soret band in ferrous STEAP2 shifts to 427 nm and the α and β bands are resolved at 560 nm and 532 nm, respectively (*Figure 4B*). We further characterized the heme using magnetic circular dichroism (MCD) spectroscopy. The MCD spectrum of ferric STEAP2 shows strong Soret signals between 404 nm and 419 nm and no high-spin charge-transfer signal at wavelength above 600 nm while ferrous STEAP2 shows a much weaker Soret band but a very strong α band from 554 nm to 562 nm (*Figure 4C*), consistent with the intense A-term Faraday effect of a typical low-spin *b*-type heme. Combined, the spectroscopic data indicates a *bis*-imidazole ligated low-spin heme in both redox states of STEAP2, consistent with a role in mediating electron transfer.

We monitored the spectral changes in STEAP2 in the reaction with NADPH. STEAP2 was preincubated with equal molar amount of FAD and reacted anaerobically with NADPH (*Figure 4—figure supplement 1A*). In this reaction, the Soret absorbance of heme shifts from that of ferric state ($A_{413}$) to that of ferrous heme ($A_{427}$) while the α and β absorptions split into well-resolved peaks at 560 nm and 532 nm, respectively, indicating the reduction of heme (*Figure 4—figure supplement 1A*). Two spectral species are resolved with a transition rate constant of 1.2 $(\pm 0.2) \times 10^{-3}$ s$^{-1}$ (*A* to *B*). Spectral species *A* corresponds to ferric STEAP2 plus FAD and spectral species *B* represents ferrous STEAP2

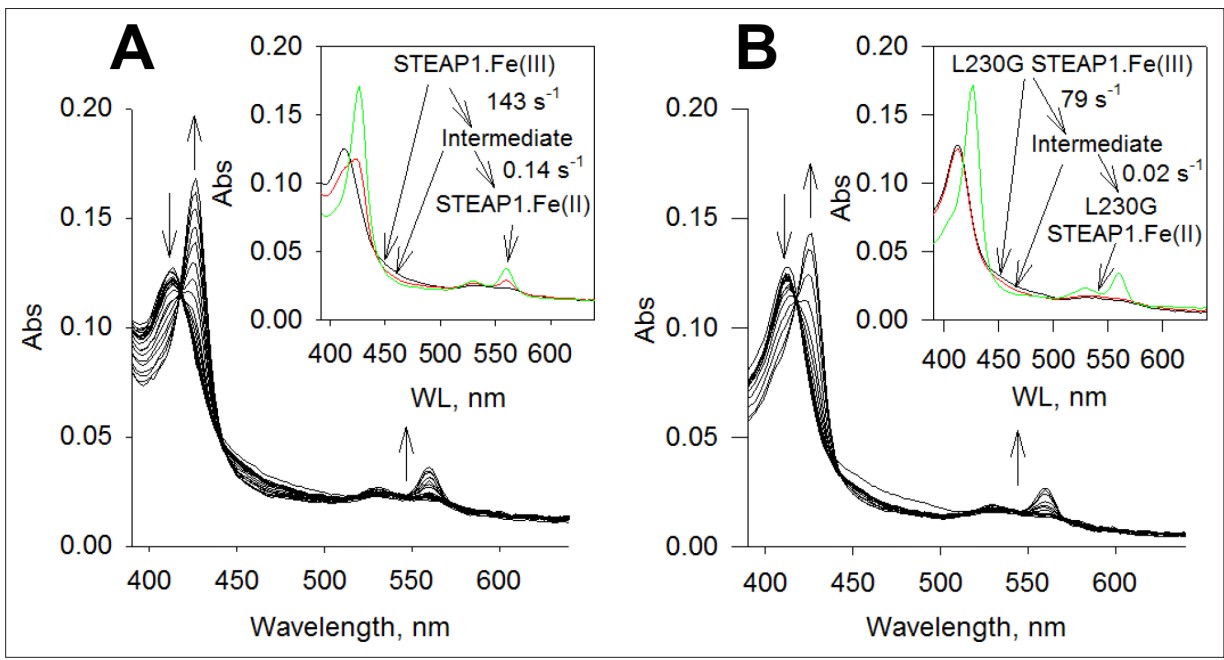

**Figure 3.** Reduction of six-transmembrane epithelial antigen of the prostate 1 (STEAP1) by cytochrome $b_5$ reductase ($b_5$R). (**A**) The rapid-scan reaction of 1.5 µM STEAP1 and 1.5 µM cytochrome $b_5$R with 10 µM NADH; the spectral change was monitored for 20 s. The arrows indicate the direction of the spectral change. Inset: the resolved spectral species and the conversion rate constants. Black, ferric STEAP1 with $b_5$R, red, a spectral intermediate, and green, ferrous STEAP1 with fully reduced $b_5$R. (**B**) L230G STEAP1 and $b_5$R were reacted with 10 µM NADH; the spectral change was monitored for 50 s. The direction of spectral change is indicated by the arrows. Inset, the resolved spectral species by deconvolution and the rate constants. Inset: black, ferric L230G STEAP1 with $b_5$R, red, a spectral intermediate, and green, ferrous L230G STEAP1 with fully reduced $b_5$R.

The online version of this article includes the following figure supplement(s) for figure 3:

**Figure supplement 1.** Six-transmembrane epithelial antigen of the prostate 1 (STEAP1) in the presence of FAD and/or NAD(P)H.

**Figure supplement 2.** Binding of cytochrome $b_5$ reductase ($b_5$R) to six-transmembrane epithelial antigen of the prostate 1 (STEAP1).

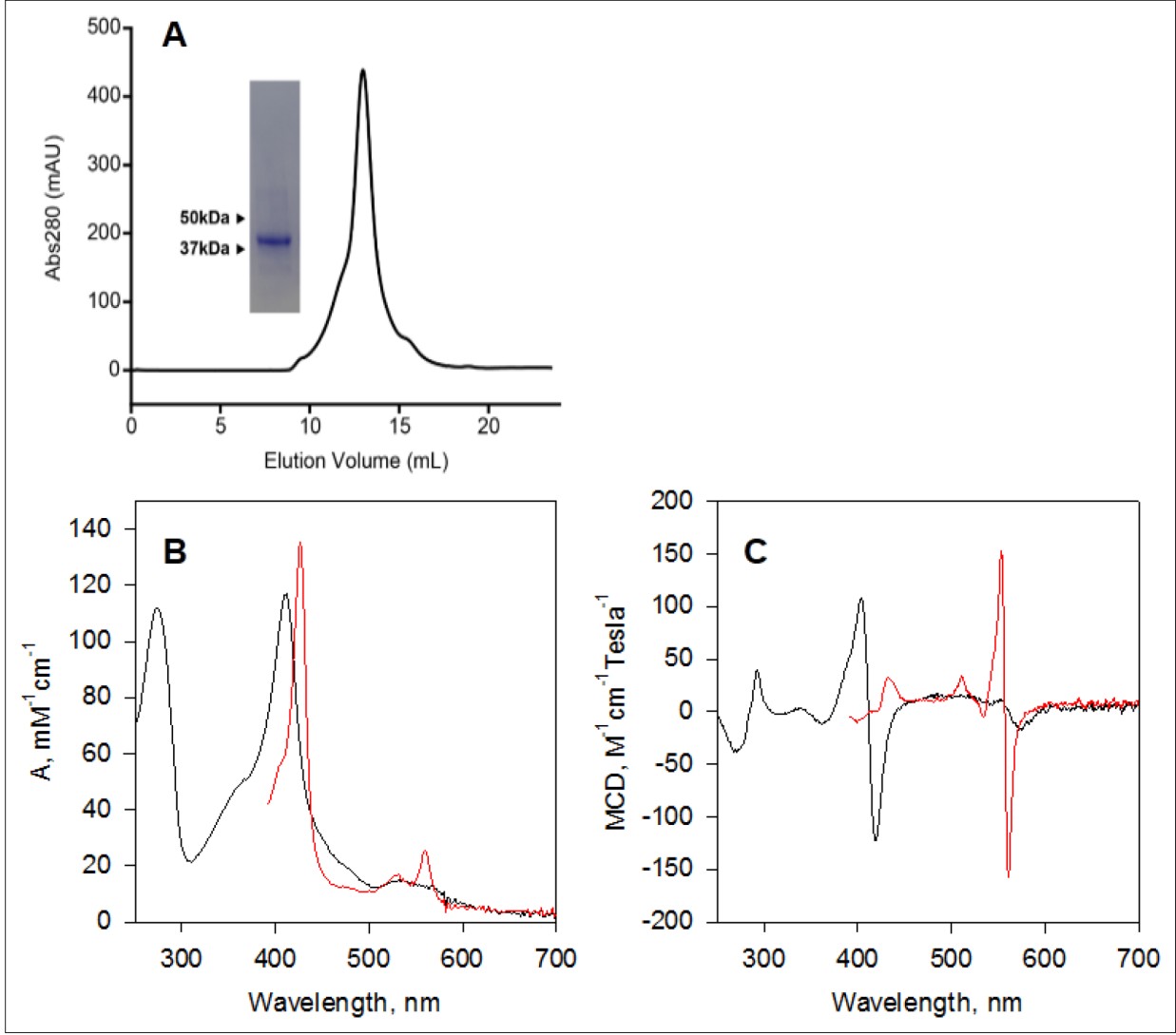

**Figure 4.** Characterization of purified human six-transmembrane epithelial antigen of the prostate 2 (hSTEAP2). (**A**) SDS-PAGE analysis and size exclusion chromatography of purified hSTEAP2. The UV-Vis (**B**) and magnetic circular dichroism (MCD) (**C**) spectra of hSTEAP2 are plotted with absorbance in mM⁻¹ cm⁻¹ and MCD in M⁻¹ cm⁻¹ Tesla⁻¹, respectively. STEAP2 was purified in the ferric state (black) and the ferrous STEAP2(red) was prepared by dithionite reduction.

The online version of this article includes the following figure supplement(s) for figure 4:

**Figure supplement 1.** Reduction of the heme on six-transmembrane epithelial antigen of the prostate 2 (STEAP2).

with reduced FAD, respectively (*Figure 4—figure supplement 1A*, inset). Under the current experimental conditions, no intermediate with reduced FAD and ferric heme is resolved, suggesting that the oxidation of FAD by heme is significantly faster than its reduction by NADPH.

## STEAP1 reduction by STEAP2

Following the characterization of STEAP2, we investigated whether the reduced FAD produced in STEAP2 is accessible to STEAP1. We prepared an anaerobic mixture of STEAP2 (pre-incubated with equal molar of FAD) and STEAP1 and then added NADPH (*Figure 2C*). In this reaction, the Soret absorbance of heme shifts from 413 nm to 427 nm and finally a narrow peak is observed at 427 nm with no shoulder at 413 nm (*Figure 2C*). This result indicates that the heme on both STEAP1 and STEAP2 is fully reduced since both isozymes have Soret absorbance at 413 nm in the ferric state and at 427 nm in the ferrous state. Two spectral species are resolved from the reaction of the STEAP mixture with NADPH, corresponding to ferric STEAP plus FAD and ferrous STEAP with reduced FAD,

with a rate constant of $5.5 \times 10^{-4}$ $s^{-1}$ (**Figure 2C**, inset). On the other hand, in the absence of STEAP2, only a small fraction of STEAP1 is reduced (**Figure 3—figure supplement 1B**), which may come from non-enzymatic reduction of FAD by NADPH. Thus, the heme on STEAP1 can be reduced by NADPH in the presence of STEAP2, via the reduced FAD produced in the OxRD of STEAP2. This data suggests that the reduced FAD becomes diffusible and quickly finds its binding pocket in the TMD of STEAP1. Under the current experimental conditions, the rate-limiting step seems to be the production of reduced FAD in the OxRD of STEAP2. Moreover, we cannot resolve the difference between diffusion of reduced FAD from STEAP2 to STEAP1 versus repositioning of reduced FAD from the OxRD to TMD in STEAP2.

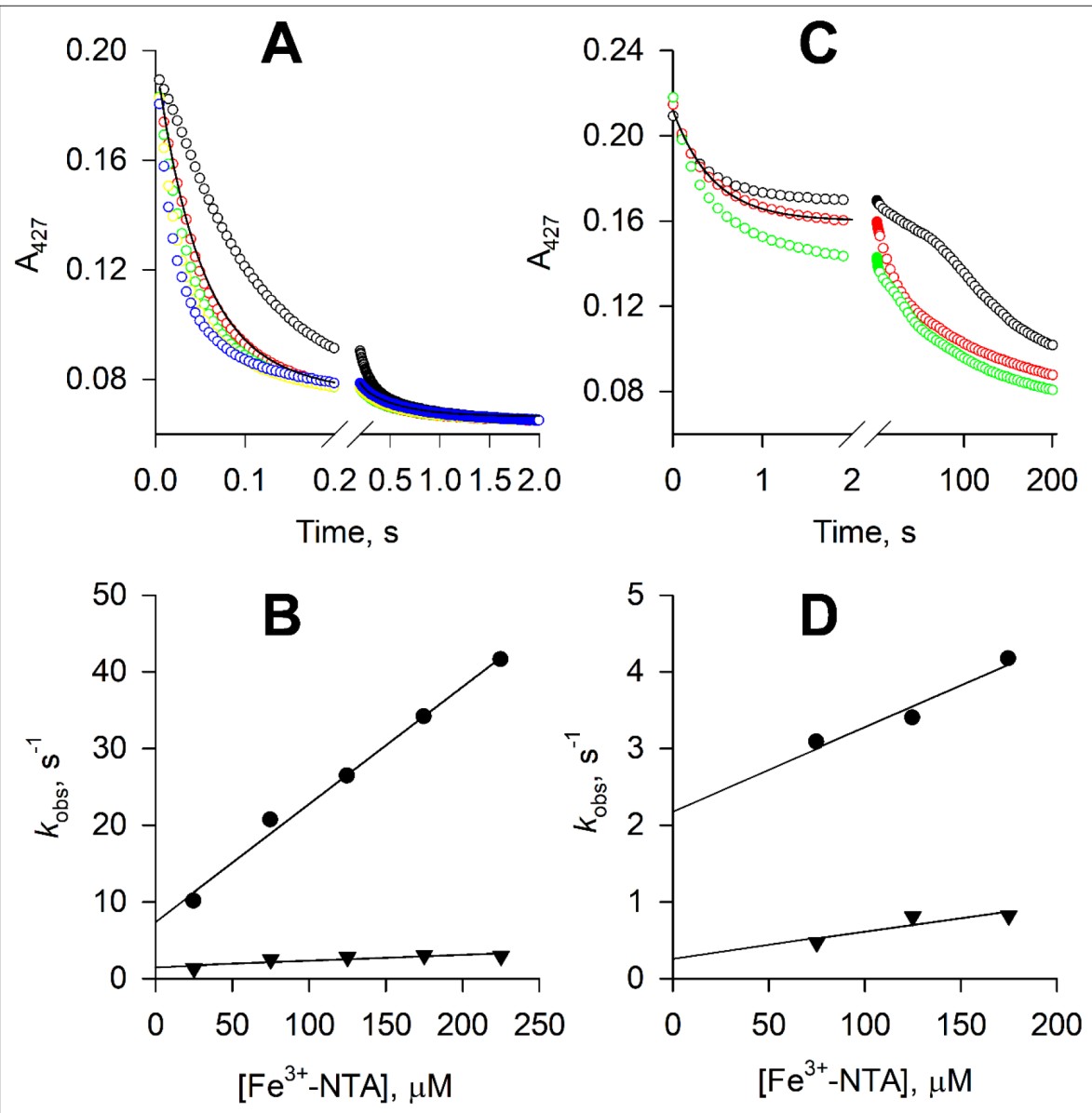

**Figure 5.** Reduction of ferric nitrilotriacetic acid ($Fe^{3+}$-NTA) by ferrous six-transmembrane epithelial antigen of the prostate 1 (STEAP1) and STEAP2. (**A**) The time courses of $A_{427}$ in the reactions of 1.1 µM ferrous STEAP1 with 25 (black), 75 (red), 125 (green), 175 (yellow), and 175 µM $Fe^{3+}$-NTA (blue). The rate constants, $k_{obs}$, are estimated by biphasic exponential fit to the time courses. One of such fits is shown by the black line. (**B**) Dependence of the rate constants $k_{obs}$ on [$Fe^{3+}$-NTA]. Circles, $k_{obs}$ of the fast phase of the $A_{427}$ time courses; triangles, $k_{obs}$ of the slow phase. (**C**) The time courses of $A_{427}$ in the reactions of 1.1 µM ferrous STEAP2 with 75 (black), 125 (red), and 175 µM $Fe^{3+}$-NTA (green). The time courses in the initial 2 s of the reactions are fitted with a biphasic exponential function, and one such fit is shown by the black line. (**D**) The rate constants estimated for the initial 2 s, $k_{obs}$, are plotted versus [$Fe^{3+}$-NTA]. Circles, $k_{obs}$ of the fast phase of the $A_{427}$ time courses; triangles, $k_{obs}$ of the slow phase. At reaction time longer than 2 s, the time courses in (**C**) show more complicated kinetics and no clear dependence on [$Fe^{3+}$-NTA].

**Table 1.** The rate constants of the reduction of ferric substrates by ferrous six-transmembrane epithelial antigen of the prostate 1 (STEAP1) and STEAP2.

| | Substrates | First phase | | Second phase | |
|---|---|---|---|---|---|
| | | $k_{on}$, $M^{-1}$ $s^{-1}$/$k_{off}$, $s^{-1}$ | $K_D$, µM* | $k_{on}$, $M^{-1}$ $s^{-1}$/$k_{off}$, $s^{-1}$ | $K_D$, µM* |
| STEAP1[†] | $Fe^{3+}$-EDTA | $2.7 \times 10^5$/4.8 | 17.8 | $4.0 \times 10^4$/1.4 | 35 |
| STEAP1[†] | $Fe^{3+}$-citrate | $1.6 \times 10^5$/15.5 | 97 | – | – |
| STEAP1[‡] | $Fe^{3+}$-NTA | $1.5 \times 10^5$/7.5 | 50 | $7.6 \times 10^3$/0.2 | 26.3 |
| STEAP2[‡, §] | $Fe^{3+}$-NTA | $1.1 \times 10^4$/2.2 | 200 | $3.5 \times 10^3$/0.3 | 85.7 |

*: $K_D = k_{off}/k_{on}$.
[†]From Kim, K. et al, *Biochemistry* (2016) **55**, 6673–6684.
[‡]This study.
[§]The third phase is not included.

## $Fe^{3+}$-NTA reduction by STEAP1 and STEAP2

Our ability to produce reduced STEAP1 and STEAP2 allows measurement of the electron transfer step between the heme and a substrate. When reduced STEAP1 was mixed with ferric nitrilotriacetic acid ($Fe^{3+}$-NTA), the $A_{427}$ absorbance decreased, indicating that the ferrous heme is oxidized by $Fe^{3+}$-NTA. The time courses of $A_{427}$ show biphasic kinetics with a fast phase accounting for 85% of the total absorbance change and a slow phase accounting for 15% (*Figure 5A*). Similar biphasic kinetics was previously observed in the reactions of ferrous STEAP1 with $Fe^{3+}$-EDTA or $Fe^{3+}$-citrate (*Kim et al., 2016*). We speculate that there are two STEAP1 populations in terms of the conformation of the substrate binding site or geometry of the heme, or both. The $k_{on}$ and $k_{off}$ rate constants of $Fe^{3+}$-NTA are estimated based on the $k_{obs}$ vs. [$Fe^{3+}$-NTA] (*Figure 5B*) and the $K_D$ values of $Fe^{3+}$-NTA for the two populations of STEAP1 are calculated, 50 µM and 26.3 µM, for the fast and slow phase, respectively (*Table 1*). The $K_D$ values of $Fe^{3+}$-NTA are similar to those of $Fe^{3+}$-EDTA or $Fe^{3+}$-citrate (*Table 1*).

The oxidation of ferrous STEAP2 by $Fe^{3+}$-NTA is significantly slower than that of STEAP1 and the time courses of $A_{427}$ show more than two phases (*Figure 5C*), suggesting more heterogeneity in the substrate binding site or heme geometry. The $\Delta A_{427}$ in the initial 2 s accounts for about half of the total $A_{427}$ decrease and the time courses in the initial 2 s can be fitted with a biphasic exponential decay function (*Figure 5C*). The rate constants $k_{obs}$ of the two phases show weak dependence on [$Fe^{3+}$-NTA] (*Figure 5D*). The $k_{on}$ and $k_{off}$ rate constants of $Fe^{3+}$-NTA binding are estimated based on the $k_{obs}$ vs. [$Fe^{3+}$-NTA] plot (*Figure 5D*) and the $K_D$'s of $Fe^{3+}$-NTA are calculated, 200 µM and 85.7 µM, for the fast and slow phases, respectively (*Table 1*). After the initial 2 s, the time courses of $A_{427}$ become slower with varying complicated shapes and have no clear dependence on [$Fe^{3+}$-NTA] (*Figure 5D*).

## Cryo-EM structure of STEAP2

To understand the structural basis for the slow $Fe^{3+}$-NTA reduction by STEAP2 compared to STEAP1, we determined the structure of STEAP2 in the presence of $NADP^+$ and FAD using cryo-EM. The cryo-EM data collection, refinement, and validation statistics are summarized in *Table 2*. The quality of density map is sufficient to build all the major structural elements of STEAP2 de novo with an overall resolution of 3.2 Å (*Figure 6—figure supplement 1*, *Figure 6—figure supplement 2*). The N-terminal residues 1–27, C-terminal residues 470–490, and residues 332–353 (loop between TM3 and -4) are not resolved in our EM map, likely due to high degree of flexibility (*Figure 6D*). Similar to the structure of STEAP4, STEAP2 has a domain-swapped homotrimer structure, where the OxRD of one protomer interacts with the TMD of a neighboring protomer (*Figure 6A and B*). Overall, the structure of STEAP2 is very similar to that of STEAP4 with a root mean squared distance of 0.8 Å (Cα) (*Oosterheert et al., 2018*). In STEAP4, the residues corresponding to 332–353 in STEAP2 form a well-defined extracellular loop adjacent to the putative substrate binding site. The high flexibility of these residues in STEAP2 may lead to poor binding of $Fe^{3+}$-NTA and its slow reduction by STEAP2.

Heme, FAD, and $NADP^+$ are unambiguously resolved in the density map (*Figure 6C* and *Figure 6—figure supplement 1*). The FAD molecule adopts an extended conformation as observed in STEAP4 (*Oosterheert et al., 2018*). The isoalloxazine ring buries deep in the TMD while the adenine ring

**Table 2.** The data collection, refinement, and validation statistics of six-transmembrane epithelial antigen of the prostate 2 (STEAP2) cryo-electron microscopy (cryo-EM).

| Data collection and processing | |
|---|---|
| Magnification | FEI Titan Krios |
| Voltage (kV) | 300 |
| Electron exposure (e⁻ Å⁻²) | 50 |
| Defocus range (μm) | –0.8 to –2.5 |
| Pixel size (Å) | 1.08 |
| Symmetry imposed | C3 |
| Number of initial particle images | 4,210,570 |
| Number of final particle images | 117,053 |
| Map resolution (Å) | 3.16 |
| FSC threshold | 0.143 |
| Map resolution range (Å) | 3.2 |
| **Refinement** | |
| Initial model used | PDB 6hcy |
| Model resolution (Å) | 3.2 |
| FSC threshold | 0.5 |
| Map sharpening B factor (Å²) | –100 |
| Model composition | |
| Non-hydrogen atoms | 11,109 |
| Protein residues | 1260 |
| Ligands | 18 |
| B factors (Å²) | |
| Protein | 41.92 |
| Ligand | 32.76 |
| R.m.s. deviations | |
| Bond lengths (Å) | 0.003 |
| Bond angles (°) | 0.567 |
| Validation | |
| MolProbity score | 1.6 |
| Clashscore | 10.54 |
| Poor rotamers (%) | 0.09 |
| Ramachandran plot | |
| Favored (%) | 97.8 |
| Allowed (%) | 2.2 |
| Disallowed (%) | 0 |

of FAD forms a stacking interaction with Trp152 from the OxRD. The ribityl and pyrophosphate in FAD also interacts with residues on TMD. The isoalloxazine ring is ~10 Å away (edge-to-edge) from the heme and the side chain of Leu371 protrudes approximately midway in between (*Figure 6E*,). Like L230 in STEAP1 (*Figure 2—figure supplement 1A*), Leu371 may mediate electron transfer in STEAP2. The distance of the isoalloxazine ring of FAD to the nearest nicotinamide ring of NADP⁺ is

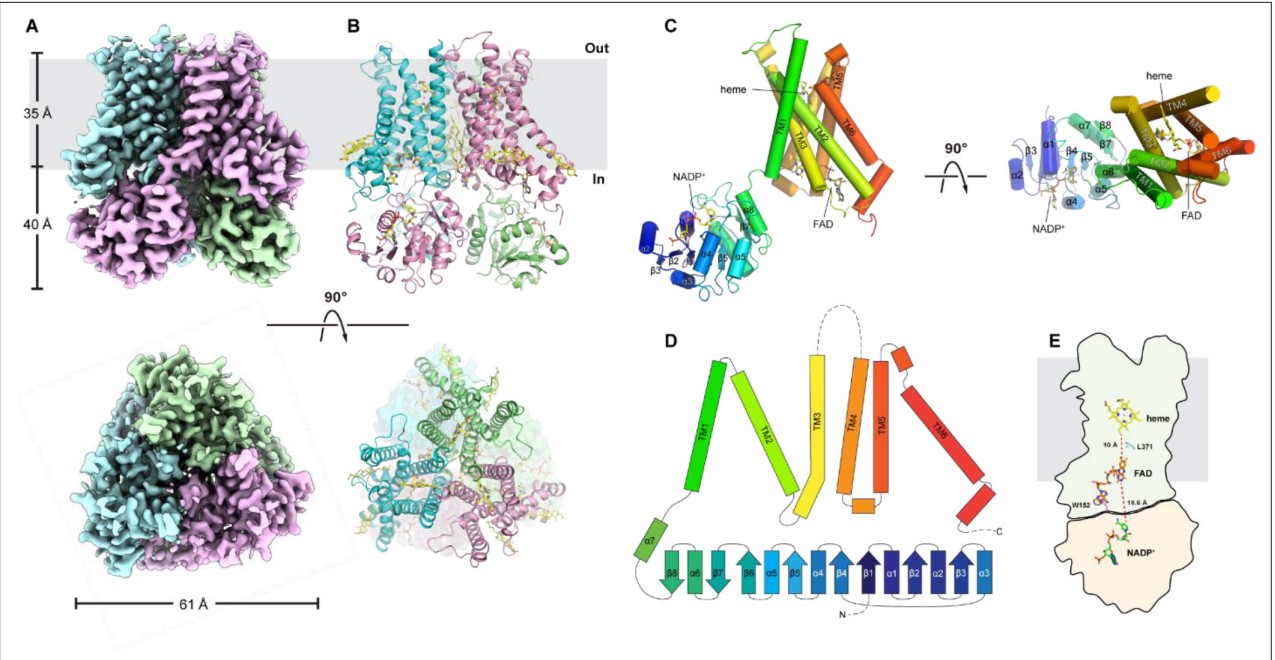

**Figure 6.** Cryo-electron microscopy (cryo-EM) structure of six-transmembrane epithelial antigen of the prostate 2 (STEAP2). The sharpened density map (**A**) and cartoon presentation (**B**) for STEAP2 homotrimer. Top, the side view of STEAP2 homotrimer, and the gray bar represents the membrane; 'in', the intracellular side and 'out', the extracellular side. Bottom, the top view of STEAP2 homotrimer from the extracellular side. (**C**) The structure of one STEAP2 protomer (cartoon) with the prosthetic group heme and the cofactors FAD and NADP⁺ (sticks). Left, side view and right, top view from the extracellular side. (**D**) The topographic representation of the secondary structural elements. The α helices and β strands are represented by bars and arrows respectively. Dashed lines represent the unresolved segments. (**E**) The schematic representation of the spatial relationship of NADP⁺, FAD, and heme, shown as sticks. Trp152 and Leu371 are also shown as sticks. Transmembrane domain (TMD) is represented as the outline with gray shade and the oxidoreductase domain (OxRD) with pink shade.

The online version of this article includes the following figure supplement(s) for figure 6:

**Figure supplement 1.** Density maps of the structural elements, heme, FAD, NADP⁺, and lipids in the cryo-electron microscopy (cryo-EM) of hSTEAP2.

**Figure supplement 2.** The images (**A**) and processing (**B–D**) of the electron microscopy (EM) data of hSTEAP2.

---

~19 Å (*Figure 6E*), which is too long for direct hydride transfer. We also found densities that likely correspond to phospholipid and cholesterol molecules. A 1-palmitroyl-2-oleoyl-glycero-3-phosphoch oline (POPC) molecule was built between the TMDs of two neighboring protomers, and two choles-terol molecules were built on the periphery of each TMD (*Figure 6—figure supplement 1*). These tightly bound lipid molecules may have relevant structural and functional roles in STEAP2.

## Reduction of STEAP2 by reduced FAD

We measured reduction STEAP2 by reduced FAD in the absence of NADPH. Reacting with 4.5 μM reduced FAD (FADH⁻), ferric STEAP2 is reduced in biphasic kinetics with rates of 2.9 (±0.80) s⁻¹ (16%) and 6.9 (±0.33)×10⁻² s⁻¹ (84%), respectively (*Figure 4—figure supplement 1B*). The reduction of STEAP2 by reduced FAD is significantly slower than that of STEAP1, and we attribute this to the presence of OxRD in STEAP2, which binds to the adenosine moiety of FAD but obstructs entrance of the isoalloxazine ring of reduced FAD into the TMD.

## Discussion

In this study, we demonstrate that STEAP1 is reduced by reduced FAD, either supplied directly or produced in the OxRD of STEAP2. We also show that $b_5$R can reduce STEAP1 in the presence of NADH. Thus, STEAP1 may partner with various flavin-dependent reductases to establish an electron transfer chain from either NADH or NADPH to the extracellular side. These discoveries will facilitate our understanding of the physiological functions of STEAP1.

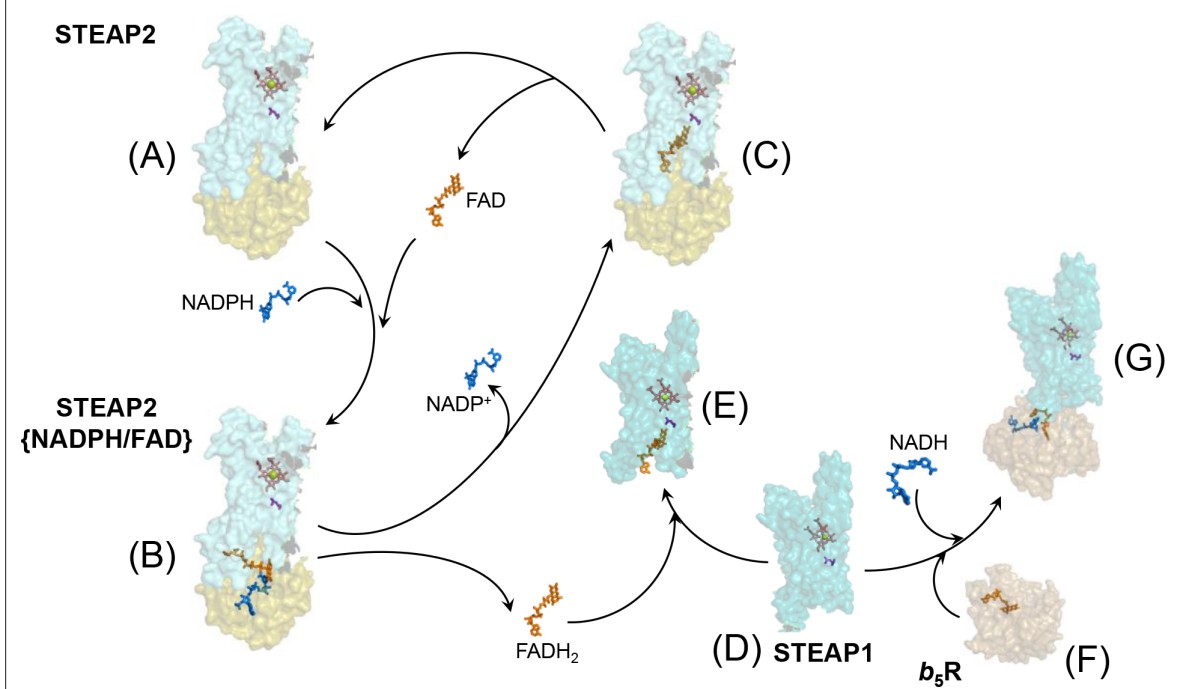

**Figure 7.** Electron transfer in six-transmembrane epithelial antigen of the prostate 1 (STEAP1) and STEAP2. NADPH (blue) and FAD (orange) bind to the oxidoreductase domain (OxRD) in STEAP2 (**A**, olive shade) with the nicotinamide ring of NADPH aligned with the isoalloxazine ring of FAD for hydride transfer (**B**). The reduced FAD adopts the extended conformation with its isoalloxazine ring bound deep in the transmembrane domain (TMD) of STEAP2 (**C**, teal shade) or dissociates from the OxRD to bind STEAP1 (**D → E**, cyan shade) and transfers electrons to heme (salmon). NADP(H) and FAD(H$_2$) are cofactors that associate with and dissociate from the STEAP protein in each redox cycle while the heme, as a prosthetic group, stays bound to the protein. Cytochrome $b_5$ reductase (**F**, sand shade) docks on STEAP1 from the intracellular side, forming a complex for electron transfer (**G**). The FAD-to-heme electron transfer in STEAP is likely mediated through a bulky side chain (purple), Leu230 in STEAP1 and Leu371 in STEAP2, respectively.

Purified STEAP2 has low levels of bound FAD, indicating that the FAD cofactor is not tightly bound as observed in other flavin reductases such as $b_5$R or P450 reductase. In the structure of STEAP2, we captured the bound FAD in a conformation suitable for transferring electrons to heme, and the structure is similar to that of STEAP4 (*Oosterheert et al., 2018*). However, the structure of STEAP2 with the bound FAD in position to receive electrons from NADPH remains unresolved. Reduction of STEAP1 in the presence of STEAP2 provides the first evidence that the bound FAD may dissociate from STEAP2 after receiving electrons from NADPH. We propose that FAD first binds to OxRD of STEAP2 in a folded conformation with its isoalloxazine ring aligned with NADPH for hydride transfer. After receiving electrons from NADPH, reduced FAD may either dissociate from the OxRD and can be utilized by STEAP2 or STEAP1 nearby or stays partially bound to the OxRD and changes into an extended conformation with the isoalloxazine ring inserted deeply into the TMD (*Figure 7*). A 'diffusible' FAD or an FAD that switches between the folded and extended conformations will inevitably limit the rate of the overall electron transfer from NADPH to the extracellular side and is consistent with the kinetics data presented in *Figure 2C* and *Figure 4—figure supplement 1A*. This mechanism is different from the electron transfer chains found in the closely related NOX or DUOX enzymes, in which the FAD likely stays bound to the OxRD and thus suitable for rapid transfer of electrons from NADPH to the extracellular side. Moreover, the mechanism of regulating the activities of STEAPs in vivo must be very different from those in NOX or DUOX (*Bedard and Krause, 2007*) and warrants further investigation.

We demonstrate that STEAP1 can establish electron transfer chain with $b_5$R, which does not release FAD. We also show that $b_5$R reduces STEAP1 from the intracellular side and it can form a complex with STEAP1 (*Figure 7*). These results indicate that $b_5$R can serve as a surrogate OxRD to complete the electron transfer chain. However, further analysis is required to establish whether STEAP1 pairs with $b_5$R or other FAD reductase in vivo.

We are able to measure the rate of electron transfer from heme to a ferric substrate, and we found that STEAP2 reduces $Fe^{3+}$-NTA significantly slower than STEAP1. The more complex time course of heme oxidation also suggests a high level of heterogeneity in either the substrate binding pocket or the heme geometry in STEAP2. In the cryo-EM structure of STEAP2, residues 332–353 between TM3 and 4 are unresolved, likely due to high flexibility in this region, while in the structures of STEAP1 and STEAP4, the corresponding residues form a well-defined extracellular loop adjacent to the putative substrate binding site (*Oosterheert et al., 2018*; *Oosterheert and Gros, 2020*). These differences suggest that STEAP2 has different substrates from STEAP4.

## Materials and methods

### Materials

4-(2-Hydroxyethyl)-1-piperazineethanesulfonic acid (HEPES), 5-aminolevulinic acid (5-ALA), isopropyl β-D-1-thiogalactopyranoside, phenylmethanesulfonyl fluoride, $Fe(NO_3)_3$, hemin chloride, dithionite, POPC, NTA, 1-ethyl-3-[3-dimethylaminopropyl]carbodiimide hydrochloride (EDC), and N-hydroxysulfosuccinimide (Sulfo-NHS) were from Sigma-Aldrich (St. Louis, MO, USA). Lauryl maltose neopentyl glycol (LMNG) and glyco-diosgenin (GDN) were from Anatrace (Maumee, OH, USA).

### Protein expression and purification

The human STEAP2 gene (NCBI accession number AAN04080.1) was codon optimized and cloned into a modified pFastBac Dual expression vector for production of baculovirus according to the Bac-to-Bac method (Thermo Fisher Scientific, Waltham, MA, USA). P3 viruses were used to infect High Five (*Trichoplusia ni*) or Sf9 (*Spodoptera frugiperda*) insect cells at a density of ~$3 \times 10^6$ cells $mL^{-1}$ in the media including 0.5 mM 5-ALA, 10 µM $FeCl_3$, and 5 µM hemin chloride. Infected cells were grown at 27 °C for 48–60 hr before harvest. Cell membranes were prepared using a hypotonic/hypertonic wash protocol as previously described (*Bai et al., 2015*). Purified cell membrane pellets were then flash-frozen in liquid nitrogen for further use.

Purified membranes were thawed and homogenized in 20 mM HEPES, pH 7.5 buffer containing 150 mM NaCl, and then solubilized with 1.5% (wt/vol) LMNG at 4 °C for 2 hr. After solubilization, cell debris was removed by ultracentrifugation (55,000×g for 45 min at 4 °C), and hSTEAP2 was purified from the supernatant using cobalt-based Talon affinity resin (Clontech, Mountain View, CA, USA). The C-terminal $His_6$-tag was cleaved with tobacco etch virus (TEV) protease at room temperature for 30 min. The protein was concentrated to around 5 mg $mL^{-1}$ using an Amicon spin concentrator with a 100 kDa cut-off (Millipore, Burlington, MA, USA), and then loaded onto an SRT-3C SEC-300 size exclusion column (Sepax Technologies, Newark, DE, USA) equilibrated with 20 mM HEPES buffer containing 150 mM NaCl and 0.01% (wt/vol) LMNG. For the sample used in the cryo-EM structural studies, the size exclusion column was equilibrated with 20 mM HEPES buffer containing 150 mM NaCl and 0.02% GDN.

Rabbit STEAP1 (NCBI accession number NP_001164745.1) was expressed and purified following the method published previously (*Kim et al., 2016*). The L230G STEAP1 mutation was introduced by the QuikChange method (Stratagene, CA, USA) using the primers:

> forward, 5'- CGTGGGACTGGCTATCGGCGCTTTGCTGGCTGTGAC-3';
> reverse, 5'- GTCACAGCCAGCAAAGCGCCGATAGCCAGTCCCACG-3'.

The cDNA of mouse cytochrome $b_5R$ (UniProt Q3TDX8, soluble form, residues 24–301) was subcloned into a pET vector which appends a polyhistidine tag and a TEV protease site to the N-terminus of the overexpressed protein. The expression of $b_5R$ followed the previous protocol (*Shen et al., 2020*) and the cell media was supplemented with 100 µM FAD.

### Electronic absorption and MCD spectroscopy

UV-Vis spectra of STEAP2 were recorded using a HP8453 diode-array spectrophotometer (Hewlett-Packard, Palo Alto, CA, USA). The extinction coefficient of the heme was determined by pyridine hemochrome assay as published previously (*Kim et al., 2016*). MCD spectra of STEAP2 were recorded with a Jasco J-815 CD spectropolarimeter (Tokyo, Japan) equipped with an Olis permanent magnet (Bogart, GA, USA). The parameters for MCD measurements are spectral bandwidth, 5 nm; time

constant, 0.5 s; scan speed, 200 nm min$^{-1}$. Each MCD spectrum is an average of 12 repetitive scans and the signal intensity is expressed in units of M$^{-1}$ cm$^{-1}$ Tesla$^{-1}$.

## Cryo-EM structure determination of STEAP2

Quantifoil R1.2/1.3 Cu grids were glow-discharged in air for 15 s at 10 mA in a plasma cleaner (PELCO EasiGlow, Ted Pella, Inc, Redding, CA, USA). Glow-discharged grids were prepared using Thermo Fisher Vitrobot Mark IV. Concentrated hSTEAP2 in the presence of FAD and NADP$^+$ (3.5 μL) was applied to each glow-discharged grid. After blotted with filter paper (Ted Pella, Inc) for 4.0 s, the grids were plunged into liquid ethane cooled with liquid nitrogen. A total of 7509 micrograph stacks were collected using SerialEM (*Mastronarde, 2005*; *Schorb et al., 2019*) on a Titan Krios electron microscope (Thermo Fisher) at 300 kV with a Quantum energy filter (Gatan, Pleasanton, CA, USA), at a nominal magnification of ×105,000 and with defocus values of −2.5 μm to −0.8 μm. A K3 Summit direct electron detector (Gatan) was paired with the microscope. Each stack was collected in the super-resolution mode with an exposing time of 0.175 s per frame for a total of 50 frames. The dose was about 50 e$^-$ Å$^{-2}$ for each stack. The stacks were motion-corrected with MotionCor2 (*Zheng et al., 2017*) and binned (2×2) so that the pixel size was 1.08 Å. Dose weighting (*Grant and Grigorieff, 2015*) was performed during motion correction, and the defocus values were estimated with Gctf (*Zhang, 2016*).

A total of 4,210,570 particles were automatically picked (RELION 3.1) (*Kimanius et al., 2016*; *Scheres, 2012*; *Scheres, 2015*) from the motion-corrected images and imported into cryoSPARC (*Punjani et al., 2017*). After two rounds of two-dimensional classification, a total of 91 classes containing 1,031,895 particles were selected. A subset of 12 classes containing 117,053 particles were selected for ab initio three-dimensional reconstruction, producing one good class with recognizable structural features and three bad classes with no distinct structural features. Both the good and bad classes were used as references in the heterogeneous refinement (cryoSPARC) and yielded a good class at 4.10 Å from 305,849 particles. Non-uniform refinement (cryoSPARC) was then performed with C3 symmetry and an adaptive solvent mask, producing a map with an overall resolution of 3.16 Å. Resolutions were estimated using the gold-standard Fourier shell correlation with a 0.143 cut-off (*Rosenthal and Henderson, 2003*) and high-resolution noise substitution (*Chen et al., 2013*). Local resolution was estimated using ResMap (*Kucukelbir et al., 2014*).

The structural model of STEAP2 was built based on the cryo-EM structure of STEAP4 (PDB ID: 6HCY) (*Oosterheert et al., 2018*), and the side chains were adjusted based on the density map. Model building was conducted in Coot (*Emsley et al., 2010*). Structural refinements were carried out in PHENIX in real space with secondary structure and geometry restraints (*Adams et al., 2010*). The EMRinger Score was calculated as described previously (*Barad et al., 2015*).

## STEAP reduction by NADPH

STEAP2, 2.3 μM, or a mixture of 1.1 μM STEAP2 and 0.9 μM STEAP1, was pre-incubated with 2.5 μM and 2.2 μM FAD in a tonometer, respectively. The solutions were made anaerobic by 5 anaerobic cycles, each with 30 s vacuum followed by argon sparging for 4.5 min. The stock solution of NADPH was made anaerobic by N$_2$ sparging. Anaerobic NADPH solution was injected into the anaerobic STEAP solution using an airtight syringe and the spectral changes were monitored at room temperature using the HP 8453 spectrophotometer. The spectral changes were deconvoluted using the Pro-Kineticist program coming with the stopped-flow machine (see below).

## Stopped-flow experiments

To measure the electron transfer rate from ferrous STEAP to ferric nitrilotriacetic acid (Fe$^{3+}$-NTA) substrate, anaerobic STEAP was first titrated to the ferrous state using dithionite and then reacted with Fe$^{3+}$-NTA on an Applied Photophysics model SX-18MV stopped-flow machine (Leatherhead, UK), which was placed in a COY anaerobic chamber (Grass Lake, MI, USA). The time course of A$_{427}$, which reflects the oxidation of ferrous STEAP, was followed. Fe$^{3+}$-NTA was prepared with ferric nitrate and NTA based on a ratio of [Fe$^{3+}$]:[NTA]=1:4. The rate constants of the redox reactions, k$_{obs}$, were obtained by fitting the time courses using a monophasic or a multiphasic exponential function. The second-order k$_{on}$ rate constants of Fe$^{3+}$-NTA to the STEAP protein were estimated from the slopes of

the linear fits to the $k_{obs}$ vs. [$Fe^{3+}$-NTA] plots. On the other hand, the $k_{off}$ rate constants were estimated based on the intercepts on ordinate of the linear fits.

The anaerobic protein mixture of STEAP1 and $b_5$R was reacted with NADH and the spectral changes were monitored using a rapid-scan diode-array accessory with the stopped-flow machine. In the reaction of STEAPs with pre-reduced FAD, FAD in 20 mM HEPES, pH 7.5 containing 150 mM NaCl and 0.1% LMNG was titrated anaerobically with dithionite. Most of the reduced FAD ($FADH_2$) was likely in its ionized form $FADH^-$ due to its pKa = 6.7. Part of the reduced FAD was re-oxidized due to the very negative potential of the FAD/$FADH^-$ pair, and the absorbance of FAD was subtracted as the background. The spectral changes were deconvoluted using the Pro-Kineticist program coming with the stopped-flow machine.

## Octet BLI

BLI assays were performed at 30°C under constant shaking at 1000 rpm using an Octet system (FortéBio, Fremont, CA, USA). STEAP1 was immobilized on amine reactive second-generation biosensors (Sartorius, Göttingen, Germany). The biosensor tips were activated for 5 min in EDC and 10 mM Sulfo-NHS before being loaded with STEAP1 at a concentration of 1 µg mL$^{-1}$ for 10 min. The tips were then quenched in 1 M ethanolamine (pH 8.5) for 5 min and equilibrated in 20 mM HEPES, pH 7.5 containing 150 mM NaCl, 0.1% LMNG, and 0.1% BSA to reduce non-specific binding. The tips were then transferred into wells containing various concentration of $b_5$R, 20, 10, 5, 2.5, 1.3, and 0.6 µM, for association and then back to the equilibration wells for dissociation. The binding curves were aligned and corrected with the channel with no analyst protein. The association and dissociation phases were fitted with a monophasic exponential function. The equilibrium responses (Req) in the association incubation were plotted against [$b_5$R] and fitted with a dose-response function to calculate the dissociation constant $K_D$ of STEAP/$b_5$R complex.

## Additional information

### Funding

| Funder | Grant reference number | Author |
|---|---|---|
| National Institutes of Health | GM145416 | Ming Zhou |
| National Institutes of Health | DK122784 | Ming Zhou |

The funders had no role in study design, data collection and interpretation, or the decision to submit the work for publication.

### Author contributions

Kehan Chen, Lie Wang, Formal analysis, Investigation, Methodology, Writing – review and editing; Jiemin Shen, Investigation, Methodology, Writing – review and editing; Ah-Lim Tsai, Resources, Formal analysis, Funding acquisition, Methodology, Writing – review and editing; Ming Zhou, Conceptualization, Resources, Data curation, Formal analysis, Supervision, Funding acquisition, Validation, Investigation, Methodology, Writing - original draft, Project administration, Writing – review and editing; Gang Wu, Conceptualization, Resources, Data curation, Formal analysis, Supervision, Validation, Investigation, Methodology, Writing - original draft, Project administration, Writing – review and editing

### Author ORCIDs

Kehan Chen http://orcid.org/0000-0003-0419-8360
Lie Wang http://orcid.org/0000-0002-9406-5168
Jiemin Shen http://orcid.org/0000-0002-3977-0681
Ming Zhou http://orcid.org/0000-0001-7198-165X
Gang Wu http://orcid.org/0000-0001-6803-0853

Reviewer #1 (Public Review): https://doi.org/10.7554/eLife.88299.3.sa1
Reviewer #2 (Public Review): https://doi.org/10.7554/eLife.88299.3.sa2

Reviewer #3 (Public Review): https://doi.org/10.7554/eLife.88299.3.sa3
Author Response https://doi.org/10.7554/eLife.88299.3.sa4

## Additional files

### Supplementary files
• MDAR checklist

### Data availability
The EM data and fitted model of human STEAP2 are deposited in the Electron Microscopy Data Bank (access code: EMD-25775): https://www.ebi.ac.uk/emdb/EMD-25775 and the RCSB Protein Data Bank (access code: 7TAI): https://www.rcsb.org/structure/7TAI. A repository of kinetics data (source data files for Figure 2 - Figure 5) has been made in Dryad: doi:10.5061/dryad.00000008r. The plasmids for protein expression are available upon request, please contact mzhou@bcm.edu or gang.wu@uth.tmc.edu.

The following datasets were generated:

| Author(s) | Year | Dataset title | Dataset URL | Database and Identifier |
|---|---|---|---|---|
| Wu G | 2023 | Kinetics of the redox reactions in STEAP1 and STEAP2 | https://doi.org/10.5061/dryad.00000008r | Dryad Digital Repository, 10.5061/dryad.00000008r |
| Wang L, Chen KH, Zhou M | 2023 | Structure of STEAP2 in complex with ligands | https://www.ebi.ac.uk/emdb/EMD-25775 | EMDataBank, EMD-25775 |
| Wang L, Chen KH, Zhou M | 2023 | Structure of STEAP2 in complex with ligands | https://www.rcsb.org/structure/7TAI | RCSB Protein Data Bank, 7TAI |

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
