## [Editor Report · eLife assessment]

This study provides **useful** insights into the mechanisms of electron transport in STEAP proteins, consistent with current models. The work strengthens and supports previously published biochemical and structural data, and the experimental results are of **solid** technical quality. The manuscript will be of interest to colleagues who work on STEAP proteins and related electron transfer systems.

---

## [Referee Report · Reviewer #1 (Public Review)]

In the revised manuscript presented by Chen, Wang, and coworkers, the authors examine two proteins, STEAP1 and STEAP2, which are transmembrane hemoproteins that are involved in Fe and Cu homeostasis and are implicated in certain cancer states. The authors produce recombinant forms of STEAP1 and STEAP2 and attempt to reconstruct the electron-transport chains of both; under certain conditions, the electron transport chain of STEAP2 consists of an internal reductase domain that binds NADPH and transfers electrons to an internal FAD molecule prior to the heme b, while STEAP1 can use an independent/external b5 reductase instead of an intrinsic reductase domain to accomplish the same electron transport pathway. A strong feature of this manuscript is the determination of the cryo-EM structure of the human STEAP2 protein resolved to 3.2 Å globally and bound to heme, FAD (in an extended conformation), and NADP+/NADPH.

This revised study aims to address the previous weaknesses that were noted, such as the unclear presentation of the kinetics data, the lack of determined redox couples, the lack of in vivo oligomerization verification, and some minor weaknesses such as the fit of the BLI data and the exact redox states of the bound coenzymes. In general, the authors have sought to rectify these weaknesses chiefly through textual edits. Through these revisions, the kinetics data are now better presented and may be more easily interpreted by the reader, how the samples for cryo-EM were prepared with the respective coenzymes is clearer, and a comparison between the oligomerization of STEAP2 and STEAP4 suggests conservation of oligomerization. The determination of the redox potentials of the hemes in both STEAP1 and STEAP2 would still be a strong addition to the data presented, but it is recognized that the limitations in the ability to prepare sufficient quantities of recombinant enzyme limits the ability to determine the measurements and may represent another publication outside of the scope of this publication.

---

## [Referee Report · Reviewer #2 (Public Review)]

Human STEAPs form a family of transmembrane heme-bound proteins. They are implicated in cancer given their high expression levels in tumor cells. Previous work has revealed that STEAPs 1-4 are iron and copper reductases. The recent structure determination of STEAP1 and STEAP4 unveiled their trimeric arrangement. STEAP1 is an outlier because it lacks the cytosolic reductase domain present in STEAPs 2-5. The present work adds to our knowledge of the family. It reports on the cryoEM structure of STEAP2 that is similar to the known structures of STEAP4 and STEAP1. The structural analysis provides additional support to a FAD-dependent heme-reduction mechanism whereby FAD oscillates between two conformations. The excellent kinetics experiments show that STEAP1 can be promiscuous regarding the source of electron donors that it can use. Indeed, cytochrome b5 can directly reduce the heme prosthetic group of STEAP1 thereby establishing an electron transfer chain that conveys electrons from NADP(P)H to the extracellular iron. Remarkably, STEAP1 can also accept electrons from free reduced FAD. Most interestingly, the manuscript demonstrates that STEAP2 can be a source of reduced FAD so that STEAP2 can create the reducing power needed for its own activity and the activity of STEAP1. This work further convincingly shows that STEAP1 can reduce iron whereas STEAP2 is less effective in iron reduction. The manuscript indicates that STEAP2 might accept other substrates providing a hint about the distinct biochemical and physiological roles of the STEAP paralogs. The manuscript does not address this point that remains open for further investigations. Aside from this minor weakness, the manuscript will advance the fields of STEAP and iron biochemistry. It has benefited from the advice given by the Reviewers leading to a high-quality presentation and data analysis.

---

## [Referee Report · Reviewer #3 (Public Review)]

The six-transmembrane epithelial antigen of the prostate (STEAP) family comprises four members in metazoans. STEAP1 was identified as integral membrane protein highly upregulated on the plasma membrane of prostate cancer cells (PMID: 10588738), and it later became evident that other STEAP proteins are also over expressed in cancers, making STEAPs potential therapeutic targets (PMID: 22804687). Functionally, STEAP2-4 are ferric and cupric reductases that are important for maintaining cellular metal uptake (PMIDs: 16227996, 16609065). The cellular function of STEAP1 remains unknown, as it cannot function as an independent metalloreductase. In the last years, structural and functional data have revealed that STEAPs form trimeric assemblies and that they transport electrons from intracellular NADPH, through membrane bound FAD and heme cofactors, to extracellular metal ions (PMIDs: 23733181, 26205815, 30337524). In addition, numerous studies (including a previous study from the senior authors) have provided strong implications for a potential metalloreductase function of STEAP1 (PMIDs: 27792302, 32409586).

This new study by Chen et al. aims to further characterize the previously established electron transport chain in STEAPs in high molecular detail through a variety of assays. This is a well-performed study that provides new insights into the established mechanism of electron transport in STEAP proteins. The authors first perform a detailed spectroscopic analysis of STEAP1, and present the interesting observation that STEAP1 can receive electrons from cytochrome b5 reductase. Then, a similar spectroscopic analysis is performed on another STEAP family member, STEAP2, followed by experiments that show how reduced FAD can diffuse from STEAP2 to STEAP1 to reduce the heme of STEAP1. Finally, the cryo-EM structure of STEAP2 is presented.

Experimentally, the conclusions are appropriate and consistent with the experimental data. The observation that STEAP1 can form an electron transfer chain with cytochrome b5 reductase in vitro is an exciting finding, but its physiological relevance remains to be validated. The metalloreductase activity of STEAP1 in vitro has been described previously by the authors and by others (PMIDs: 27792302, 32409586). However, when over expressed in HEK cells, STEAP1 by itself does not display metal ion reductase activity (PMID: 16609065), and it was also found that STEAP1 over expression does not impact iron uptake and reduction in Ewing's sarcoma (cancer) cells (PMID: 22080479). Therefore, the physiological relevance of metal ion reduction by STEAP1 remains controversial. Future studies will have to elucidate if the established interaction between STEAP1 and cytochrome b5 reductase is relevant in cells.

The work will be interesting for scientists working within the STEAP field and for those working on other oxidoreductases. The spectroscopic data is robust and However, the new structural insights into STEAP2 are limited because the structure is virtually identical to the published structures of STEAP4 and STEAP1 (PMIDs: 30337524, 32409586), which is not surprising because of the high sequence similarity between the STEAP isoforms. When taken together, this study by Chen et al. strengthens and supports previously published biochemical and structural data on STEAP proteins, making an important contribution to the STEAP field.

---

## [Author Response]

The following is the authors’ response to the original reviews.

We appreciate the critical review of our manuscript. We believe that we have addressed the questions and concerns raised by the reviewers to the best of our ability. As part of the revision, we conducted two new experiments to enhance the rigor of the conclusions and to provide more insights into the mechanism of STEAP proteins, and we reorganized the Results section, as suggested by the reviewers, following to a clearer logical thread. The new data are briefly summarized below.

1. Reduction of L230G STEAP1 by reduced FAD. We made Leu230Gly STEAP1 mutant and measured the rate of heme reduction by reduced FAD. We found that the rate of heme reduction in L230G STEAP1 is slower than that in the wild type STEAP1. Since Leu230 is solvent accessible only from the intracellular side, this result supports the conclusion that reduced FAD binds to STEAP1 on the intracellular side and reduces the heme. This result also indicates that leucine, which is found at the equivalent position in STEAP1, 2 and 3, and Phe359 in STEAP4, has a significant role in mediating electron transfer from FAD to the bound heme.

2. Reduction of STEAP2 by reduced FAD. We showed that STEAP2 can be reduced when supplied with reduced FAD, and that the rate of heme reduction is significantly slower than that of reduction of STEAP1 by reduced FAD. This result is consistent with presence of the oxidoreductase domain (OxRD)† in STEAP2, which hampers direct entrance of the isoalloxazine ring of FAD to its binding pocket in the transmembrane domain (TMD). On the other hand, the rate of heme reduction by reduced FAD is much faster than that of heme reduction in the presence of NADPH and FAD, indicating that reduction of FAD by NADPH is rate-limiting in the electron transfer chain in STEAP2.

†: To be consistent with literature, we adopted the nomenclature “oxidoreductase domain (OxRD)” for the N-terminal soluble domain in STEAP proteins. We used the term“reductase domain (RED)” in the previous version of our manuscript.

**Reviewer #1 (Public Review):**
This important study reveals the structure of human STEAP2 for the first time and suggests the electron transport pathway, but some questions remain regarding the interpretation of the in vitro electron transport experiments, the lack of available redox couples, and potential alternative hypotheses that would if addressed, strengthen the claims in the manuscript.StrengthsOne of the clear strengths of the manuscript that stands out is the determination of the structure of human STEAP2. The structures of some other homologs are known, but STEAP2's structure was not, and STEAP2 seems to have an unusually low activity towards certain metal chelates. The approach of producing the human STEAP2 in insect cells with the supplementation of cofactor biogenesis components nicely results in cofactor-replete protein. The structure of STEAP2 reveals a domain-swapped trimer, with the NADPH-binding domain of the neighboring protomer within electron-transport distance of the FAD-heme axis. The FAD has an interesting and somewhat unusual extended conformation and abuts a Leu residue that may regulate electron transport. Another strength of the manuscript is the demonstration that STEAP1, which does not have the internal NADPH binding domain, can interact modestly and shuttle electrons to the heme in STEAP1 through FAD. These experiments nicely expand information on the function of STEAP1 and provide a structural basis for electron transport in STEAP2.WeaknessesA major weakness in the manuscript lies with the kinetics data and how the data, as presented, are unclear to the reader regarding their impact and their connection to the purported electron transport scheme. While multiple sets of data are reported, the analysis in all cases is simply the reduction of a hexacoordinate heme and its related spectra and kinetic parameters. In most cases, it's unclear to the reader which part of the electron pathway is being tested in which experiment. Simple diagrams would be helpful in each case. However, it's also unclear if all of the potential order of addition experiments were actually performed; i.e., flavin but no NADPH; NADPH but no flavin; flavin before NADPH; flavin after NADPH, etc. As there are multiple permutations that should be tested to demonstrate the electron transport pathway, presenting the data in a way that makes it clear to the reader is challenging. Particularly missing are the determined redox potentials of the hemes in both STEAP1 and STEAP2. Could differences in these heme redox potentials be drivers of the difference in metal reduction rates?

We re-structured the manuscript to follow a clearer logical thread. We provided explanations for which electron transfer steps are being examined in each experiment.

We cannot reliably determine EM due to insufficient amount of purified proteins. We are inclined to think that the bound heme on STEAP1 and STEAP2 have similar EM, based on their similar coordination geometry and nearly identical UV-Vis and MCD spectra. Thus, different rates of Fe3+-NTA reduction by STEAP1 and STEAP2 are likely due to differences in substrate binding site rather than different EM.

Also, the text indicates that STEAP2 does not show a reduction rate dependence on the [Fe3+NTA], but Figure 1A shows a difference in rates dependent on [Fe3+-NTA], and the shape of the reduction curve also changes based on [Fe3+-NTA]. This discrepancy should be rectified.

We fixed this error. The reduction of Fe3+-NTA by ferrous STEAP2 shows multiple phases and the reaction rates within the initial 2 seconds are weakly dependent on [Fe3+-NTA].

A second major weakness is the lack of any verification of the relevance of the STEAP2 oligomerization to its in vivo function. Is the same domain-swapped trimer known to exist in vivo? If the protein were prepared in other detergents, is the oligomerization preserved? It is alluded to in the text that another STEAP protein is also a trimer. Was this oligomerization verified in vivo?

The domain-swapped assembly is an interesting phenomenon, and it seems to provide a solution for bringing the FAD closer to heme. The same domain swapped trimeric assembly is also observed in the structure of STEAP4, which was purified in a different detergent (Nat Commun (2018), 9, page 4337). It is likely that this feature is shared by STEAP2, 3, and 4, and preserved during the purification process.

Could this oligomerization be disrupted to impede or abrogate electron transport to underscore the oligomerization relevance? This point is germane, as it would further suggest that the domain-swapped trimer observed in the STEAP2 cryo-EM structure is physiologically relevant, especially given how far the distance between the NADPH and the FAD would otherwise be to support electron transport.

We agree with the reviewer’s reasoning that the oligomeric assembly is required for proper function of STEAPs and thus could potentially be utilized for functional regulation. However, we are not aware of any physiologically relevant stimuli or treatment that would allow regulation of STEAP functions by inducing or forming different oligomeric states. Our experience with STEAP proteins is that the trimeric assembly is stable and well-preserved during the purification process and can only be disrupted under denaturing conditions such as SDS-PAGE.

Beyond these two areas in which the manuscript could be improved there are a couple of minor considerations. First, the modest resolution of the STEAP2 structure prevents assigning the states of NADP+/NADPH and FAD/FADH2 with confidence. An orthogonal measure would be useful for modeling the accurate states in the structure.

We agree. We clarified the ambiguity and stated in the main text that the cryo-EM structure of STEAP2 was determined in the presence of NADP+ and FAD.

Finally, the BLI b5R/STEAP1 binding/unbinding fits seem somewhat poor, especially at high concentrations of b5R in the dissociation regime, which likely influences the derived value of Kd. A different fitting equilibrium might yield better agreement between the experimental and theoretical results. Moreover, whether this binding strength is influenced by the reduction state of the NADPH would be helpful in understanding and contextualizing the weak binding affinity.

We think that non-specific binding likely causes deviations from the simple binding model at higher b5R concentrations. We made a comment on this in the main text. We agree with the reviewer that the interactions between b5R and STEAP1 could be redox dependent, for example, a reduced FAD on b5R may enhance the affinity. We could implement this by performing the binding experiments in an anaerobic chamber, but this is beyond the scope of the current study.

**Reviewer #2 (Public Review):**
The manuscript provides new insight into a family of human enzymes. It demonstrates that STEAP2 can reduce iron and STEAP1 can be promiscuous regarding the source of electron donors that it can use. The quality of the kinetics experiment and the structural analysis is excellent. I am less enthusiastic about the interpretation of data and the take-home message that the manuscript intends to deliver. Above all, the work combines data on STEAP2 and STEAP1 and it remains unclear which questions are ultimately addressed. A second critical point is about the interpretation of the experiment demonstrating that STEAP1 can be reduced by cytochrome b5 reductase. The abstract states that "We show that STEAP1 can form an electron transfer chain with cytochrome b5 reductase" whereas the main text discusses the data by suggesting that "we speculate that FAD on b5R may partially dissociate to straddle between the two proteins". The two statements seem to be partly contradictory. Cytochrome b5 reductases do not easily release FAD but rather directly donate electrons to heme-protein acceptors (PMID: 36441026). According to the methods section, no FAD was added to the reaction mix used for the cytochrome b5 reductase experiment. Overall, the data seem to indicate that the reductase might reduce the heme of STEAP1 directly. Would it be possible to check whether FAD addition affects the kinetics of the process?

We agree with the reviewer on this point. We do not have evidence indicating that FAD fully or partially dissociates from b5R to interact with STEAP1. We removed the statement in the revision.

We have not tried to add free reduced FAD to the mixture of STEAP1/b5R/NADH, because we feel that this would increase the complexity of the system and complicate data interpretation. We are working on determining the structure of b5R in complex with STEAP1 to visualize the electron transfer pathway, and we hope that such a structure would provide a framework for understanding electron transfer between the two proteins.

And to perform a control experiment to check that NAD(P)H does not directly reduce the heme of STEAP1 (though unlikely)?

We did the control experiment and included data in Fig. S3A. No reduction of heme by NADH alone.

A final point concerns the "slow Fe3+-NTA reduction by STEAP2". The reaction is not that slow as the initial phase is 2 per second. The reaction does not show dependence on the substrate concentration suggesting "poor substrate binding". I am not convinced by this interpretation. Poor substrate binding would give rise to substrate dependency as saturation would not be achieved. A possible interpretation could be that substrate binding is instead tight and the enzyme is saturated by the substrate. Can it be that the reaction is limited by non-productive substrate-binding and/or by interconversions between active and non-active conformations? We re-analyzed the data and revised the Results and Discussion.

We agree with the reviewer on this point. We re-analyzed the data and found that the reaction rates within the first 2 seconds are weakly dependent on [Fe3+-NTA] while the rates beyond 2 seconds do not show dependence on [Fe3+-NTA]. More studies are required to unravel the mechanism that leads to the complicated kinetic data.

**Reviewer #3 (Public Review):**
The six-transmembrane epithelial antigen of the prostate (STEAP) family comprises four members in metazoans. STEAP1 was identified as integral membrane protein highly upregulated on the plasma membrane of prostate cancer cells (PMID: 10588738), and it later became evident that other STEAP proteins are also over expressed in cancers, making STEAPs potential therapeutic targets (PMID: 22804687). Functionally, STEAP2-4 are ferric and cupric reductases that are important for maintaining cellular metal uptake (PMIDs: 16227996, 16609065). The cellular function of STEAP1 remains unknown, as it cannot function as an independent metalloreductase. In the last years, structural and functional data have revealed that STEAPs form trimeric assemblies and that they transport electrons from intracellular NADPH, through membrane bound FAD and heme cofactors, to extracellular metal ions (PMIDs: 23733181, 26205815, 30337524). In addition, numerous studies (including a previous study from the senior authors) have provided strong implications for a potential metalloreductase function of STEAP1 (PMIDs: 27792302, 32409586).This new study by Chen et al. aims to further characterize the previously established electron transport chain in STEAPs in high molecular detail through a variety of assays. This is a wellperformed, highly specialized study that provides some useful extra insights into the established mechanism of electron transport in STEAP proteins. The authors first perform a detailed spectroscopic analysis of Fe3+-NTA reduction by STEAP2 and STEAP1, confirming that both purified proteins are capable of reducing metal ions. A cryo-EM structure of STEAP2 is also presented. It is then established that STEAP1 can receive electrons from cytochrome b5 reductase, and the authors provide experimental evidence that the flavin in STEAP proteins becomes diffusible.The specific aims of the study are clear, but it is not always obvious why certain experiments are performed only on STEAP2, on STEAP1, or on both isoforms. A better justification of the performed experiments through connecting paragraphs and proper referencing of the literature would improve readability of the manuscript. Experimentally, the conclusions are appropriate and mostly consistent with the experimental data, although one important finding can benefit from further clarification. Namely, the observation that STEAP1 can form an electron transfer chain with cytochrome b5 reductase in vitro is an exciting finding, but its physiological relevance remains to be validated. The metalloreductase activity of STEAP1 in vitro has been described previously by the authors and by others (PMIDs: 27792302, 32409586). However, when over expressed in HEK cells, STEAP1 by itself does not display metal ion reductase activity (PMID: 16609065), and it was also found that STEAP1 over expression does not impact iron uptake and reduction in Ewing's sarcoma (cancer) cells (PMID: 22080479). Therefore, the physiological relevance of metal ion reduction by STEAP1 remains controversial. The current work establishes an electron transfer chain between STEAP1 and cytochrome b5 reductase in vitro with purified proteins. However, the conformation of this metalloreductase activity of the STEAP1-cytochrome b5 complex will be required in a cell line to prove that the two proteins indeed form a physiological relevant complex and that the results are not just an in vitro artefact from using high concentrations of purified proteins.The work will be interesting for scientists working within the STEAP field. However, some of the presented data are redundant with previous findings, moderating the study's impact. For instance, the new structural insights into STEAP2 are limited because the structure is virtually identical to the published structures of STEAP4 and STEAP1 (PMIDs: 30337524, 32409586), which is not surprising because of the high sequence similarity between the STEAP isoforms. Moreover, the authors provide experimental evidence to prove the previous hypothesis (PMID: 30337524) that the flavin in STEAP proteins becomes diffusible, but the molecular arrangement of a STEAP protein, in which the flavin interacts with NADPH, remains unknown. Based on the manuscript title, I would also expect the in-depth characterization of STEAP1/STEAP2 hetero trimers (first identified by the authors), but this is only briefly mentioned. When taken together, this study by Chen et al. strengthens and supports previously published biochemical and structural data on STEAP proteins, without revealing many prominent conceptual advances.

We thank the reviewer for information and the broader context. We have revised the manuscript to have a clearer logical thread.

**Reviewer #1 (Recommendations For The Authors):**

Please see the "Public Review" for recommendations.

**Reviewer #2 (Recommendations For The Authors):**
Specific suggestions-The introduction should more clearly state which questions are being addressed and why STEAP1 and STEAP2 are investigated.

We have revised the Introduction to make that clearer.

-The manuscript should discuss more extensively and provide possible explanations for the substrate-independent kinetics of iron-reduction by STEAP2.

We re-analyzed the data and found the rate constants of the reactions before 2 s are weakly [Fe3+NTA]-dependent. We ascribe the weak [Fe3+-NTA]-dependence to the partial rate-limiting by substrate binding. However, we do not have a good interpretation for the reaction kinetics after 2 s which does not show [Fe3+-NTA]-dependence.

-"The rate of STEAP1(Fe(II)) oxidation by Fe3+-NTA is similar to those by Fe3+-EDTA or Fe3+-citrate, but the rates are significantly faster than STEAP2(Fe(II)) re-oxidation by Fe3+NTA (Fig. 1B)." The rates for STEAP1 should be given to substantiate this statement.

We added Table S1 in the supplementary materials that includes the 2nd order association (kon) and the 1st order dissociation rate constants (koff) of iron substrates in STEAP1 and STEAP2. Data on Fe3+-EDTA or Fe3+-citrate by STEAP1 are from our previous study (Biochemistry, 2016). We also calculated the KDs of different iron substrates for STEAP1 and STEAP2.

"Our results indicate that STEAP2 can supply reduce FAD to initiate electron transfer inSTEAP1." As discussed above, this statement should be discussed and analyzed

We mixed 0.9 μM STEAP1, 1.1 μM STEAP2, and 2.2 μM FAD and added 60 μM NADPH to the system and found that the heme on both STEAP1 and STEAP2 are reduced. Since adding NADPH to STEAP1 plus FAD alone does not reduce the heme (Fig. S3B), we reasoned that reduction of the heme on STEAP1 is achieved by the reduced FAD produced on STEAP2. The reduced FAD likely dissociates from STEAP2 and then bind to STEAP1.

-Experiments on "STEAP1 reduction by STEAP2" The experiments show that "STEAP2 can supply reduce FAD to initiate electron transfer in STEAP1." Could these results be explained by heterotrimer formation in agreement with the previous data published by the authors?

In our experience, STEAP1 and STEAP2 homotrimers are stable and do not form heterotrimers when mixed. STEAP1/2 heterotrimers form only when the two proteins are co-expressed in cells (Biochemistry (2016) 55, 6673-6684).

**Reviewer #3 (Recommendations For The Authors):**
Major points:1. As a very general point: the order in which the results are presented could be greatly improved to increase the readability for non-experts. To elaborate: The manuscript starts with thespectroscopic characterization of STEAP2, then suddenly the reductase activities of STEAP1 and STEAP2 are compared; subsequently, experiments are described involving STEAP1 and cytochrome b5 reductase; then the cryo-EM structure of STEAP2 is presented etc. As a non-expert reader, this presentation of the results is confusing, especially because the paragraphs are not always connected well, and there is a lot of switching between STEAP1 and STEAP2 data. A more logical order would be to first present the STEAP2 spectroscopy and structural data, then write a connecting paragraph on why it is important to also study the electron transfer chain in STEAP1, followed by the comparison of the STEAP isoforms and the data on STEAP1 alone. The authors should include sentences on why they performed each experiment. For example, why did they determine the structure of STEAP2. What were they after that they could not retrieve from the homologous STEAP4 and STEAP1 structures? Justification of the performed experiments will make it easier for the reader, and will establish a better connection between the paragraphs.

We reorganized the data presentation in Results per the reviewer’s suggestions.

1. The physiological relevance of metal ion reduction by STEAP1 remains controversial. Because the current work establishes an electron transfer chain between STEAP1 and cytochrome b5 reductase, could the authors perform an easy experiment where they over express both STEAP1 and cytochrome b5 reductase in a cell line? If such an experiment would reveal STEAP1-dependent metal-ion reduction, it would greatly improve this part of the manuscript. If no activity is observed, the established electron transfer chain could just represent an in vitro artifact from using high concentrations of purified proteins.

This is an excellent point. We are not set up to perform the proposed experiment but will do so in the future.

1. The manuscript states that metal ion reduction of purified STEAP2 is slow, and the authors explain this by the absence of density for the extracellular region between helices 3 and 4 that are present in the structures of STEAP4 and STEAP1, resulting in a less-well defined substratebinding site. Can the authors exclude that the less-well defined substrate-binding site is a result of the detergent extraction of STEAP2? Would it be possible to measure the reductase activity of STEAP2 in purified membranes?

Detergent mostly interacts with the transmembrane domains and since the TMD region of STEAP2 aligns well with those of STEAP1 and STEAP4, we do not think that the disordered substrate binding region in STEAP2 is a consequence of detergent solubilization. It is difficult to conduct pre-steady state kinetic experiments using STEAP2 in membrane fractions.

1. The manuscript would greatly benefit from citing the literature more comprehensively to acknowledge insightful findings from authors in the field; for example, the important work by the Lawrence lab from 2015 (PMID: 26205815), which biochemically proved that STEAPs bind a single heme and that FAD bridges the TMD and RED, is not cited. The authors also mention that STEAP proteins belong to the same family as NOX proteins and cite some NOX structure papers. However, they fail to cite the first NOX structure paper (PMID: 28607049), as well the manuscript that structurally compares NOXs and STEAPs (PMID: 32815713). Similarly, the authors use SerialEM for their cryo-EM data collection but cite an old paper instead of the more recent (and relevant) SerialEM publication (PMID: 31086343).

We agree and added the references.

1. Generally, the data presented in the manuscript appear of good technical quality. However, a 'Table 1' with all relevant cryo-EM data collection and refinement statistics is completely missing as far as I can see. The authors should definitely add this to allow for the judgement of structural data quality. Without it, the manuscript is not suitable for publication.

We added Table S2 that includes relevant cryo-EM statistics.

Minor points:1. The authors write in the abstract: 'STEAP2 - 4, but not STEAP1, have an intracellular domain that binds to NADPH and FAD'. This is not correct, because it has clearly been established that FAD also majorly binds to the transmembrane domain (this is even shown by the authors in the current manuscript as well).

Agree, we corrected that in the revision.

1. Sentence from the abstract and introduction state: 'It is also unclear whether STEAP1 has metal ion reductase activity' and 'it is unclear whether STEAP1 can form a competent electron transfer chain from NADPH'. The authors should definitely add "physiologically relevant" to these sentences. Namely, the senior authors themselves concluded in their 2016 Biochemistry paper (PMID: 27792302) that STEAP1 is capable of reducing metal ion complexes. Further indications that the transmembrane domain of STEAP1 displays metalloreductase activity was published by the Gros lab (PMID: 32409586), and it was also shown that fusing the RED of STEAP4 to the TMD of STEAP1 yields a functional protein in cells that reduces metal ions.

Good point and we revised the text and included the references.

1. Why is scheme 1 not just a summarizing figure?

We could change Scheme 1 to a Figure if required by the copy editor.

1. What is the purpose of Fig. 6? A larger depiction of Fig. 5e would likely be more relevant and should be considered as a replacement. Alternatively, the structure of STEAP1 (pdb 6y9b) could be shown in combination with Fig. 7, as the mutation is performed in STEAP1.

We agree and made changes to the structural figures to enhance clarity.

1. The manuscript now contains many, single panel figures. Certain main figures could easily be combined, for example, Fig. 1 and 2 and/or Fig. 3 and 4.

We agree and made changes to reduce single panel figures.

1. In Fig. 2, 3 and 4, the spectra show changes in peak heights as a result of the ferric to ferrous heme transition. However, a time component is missing in the legend. How long do these transitions take?

We added the reaction times to the figure legends.

1. The last part of the discussion states: 'The assembly of an intracellular RED with a membrane-embedded TMD observed in NOX, DUOX, and STEAPs naturally led to the notion that NADPH, FAD, and heme form an uninterrupted rigid electron-transfer chain that shuttles electron from the intracellular cellular NADPH to the extracellular substrates. While this may be true for NOX and DUOX, in which rapid supply of electrons to their extracellular substrates are essential to their biological functions, it may not apply similarly to STEAPs since it has only one heme in the TMD, and their electron transfer relies on shuttling of FAD.' The authors should mention here that the activity of NOX and DUOX is tightly regulated by accessory proteins, Ca2+ etc. Similarly, do the authors expect that the large distance between NADPH and FAD in the structures could represent a way to regulate/dampen the metal ion reduction rates of STEAPs in vivo?

We agree. We mentioned the regulation of NOX and DUOX in Discussion. We remain puzzled by the large distance between NADPH and FAD in STEAPs and are in pursuit of a structure in which the two cofactors are “in touch” for electron transfer.